# The Use of Kappa Free Light Chains to Diagnose Multiple Sclerosis

**DOI:** 10.3390/medicina58111512

**Published:** 2022-10-24

**Authors:** Borros Arneth, Jörg Kraus

**Affiliations:** 1Institute of Laboratory Medicine and Pathobiochemistry, Molecular Diagnostics, Justus Liebig University, Feulgenstr. 12, 35392 Giessen, Germany; 2Department of Laboratory Medicine, Paracelsus Medical University and Salzburger Landeskliniken, Strubergasse 21, 5020 Salzburg, Austria; 3Department of Neurology, Medical Faculty, Heinrich-Heine-University Düsseldorf, Bergische Landstraße 2, 40629 Düsseldorf, Germany

**Keywords:** multiple sclerosis, clinical isolated syndrome, free light chains, kappa free light chains

## Abstract

*Background*: The positive implications of using free light chains in diagnosing multiple sclerosis have increasingly gained considerable interest in medical research and the scientific community. It is often presumed that free light chains, particularly kappa and lambda free light chains, are of practical use and are associated with a higher probability of obtaining positive results compared to oligoclonal bands. The primary purpose of the current paper was to conduct a systematic review to assess the up-to-date methods for diagnosing multiple sclerosis using kappa and lambda free light chains. *Method*: An organized literature search was performed across four electronic sources, including Google Scholar, Web of Science, Embase, and MEDLINE. The sources analyzed in this systematic review and meta-analysis comprise randomized clinical trials, prospective cohort studies, retrospective studies, controlled clinical trials, and systematic reviews. *Results*: The review contains 116 reports that includes 1204 participants. The final selection includes a vast array of preexisting literature concerning the study topic: 35 randomized clinical trials, 21 prospective cohort studies, 19 retrospective studies, 22 controlled clinical trials, and 13 systematic reviews. *Discussion*: The incorporated literature sources provided integral insights into the benefits of free light chain diagnostics for multiple sclerosis. It was also evident that the use of free light chains in the diagnosis of clinically isolated syndrome (CIS) and multiple sclerosis is relatively fast and inexpensive in comparison to other conventional state-of-the-art diagnostic methods, e.g., using oligoclonal bands (OCBs).

## 1. Introduction

Within the past decade, a vast array of papers and empirical sources have attempted to offer an inclusive overview of the use of immunoglobulin free light chains in cerebrospinal fluid (CSF) to aid in the diagnosis of novel multiple sclerosis. Multiple sclerosis is a typical neuroinflammatory and neurodegenerative condition associated with the central nervous system (CNS) [1]. Its etiology at this point is unclear; however, the significant pathology involves autoimmune multifocal myelin obliteration across the entirety of the CNS.

Intrathecal kappa-Free Light Chains (κ-FLC) synthesis has similar diagnostic accuracy to the well-established method of CSF-restricted oligoclonal bands (OCB) to identify patients with Multiple Sclerosis (MS), and recent studies even report its value for the prediction of early MS disease activity. Furthermore, detection of κ-FLC has significant methodological advantages in comparison to OCB detection.

Expeditious and precise diagnosis is especially important for the clinical management of patients. Early disease diagnosis is crucial because disease-modifying treatments are most effective in the beginning phase of the illness [2,3]. Therefore, an ideal biomarker should allow early diagnosis of the disease, help establish its prognosis, and be quickly and effectively verifiable. At present, there is no single specific diagnostic test for multiple sclerosis. Based on the insights of a recent survey, the preexisting diagnostic procedure for multiple sclerosis depends on clinical signs, Magnetic Resonance Imaging (MRI), and laboratory CSF testing. Until now, the diagnostic criteria for MS have not included FLCs. However, in the future, κ-FLC testing could possibly support or even replace OCB testing in CSF [4].

According to Bohle et al. [5], the primary manifestations of multiple sclerosis involve cellular abnormalities and the humoral immune system. Regardless, it is widely known that the joint actions of B and T cells play an enormous role in the overall advancement of demyelination and the generation of immunoglobulin [5]. Accordingly, in most patients, a heightened degree of immunoglobulin production in the intrathecal space is noticeable [6], and oligoclonal Immunoglobulin G( IgG) is distinguishable in the CSF. Thus, the application of free light chains in the clinical diagnosis of multiple sclerosis has been widely proposed and medically investigated by numerous researchers and scholars, especially within the last half-decade. Recent studies have investigated the efficiency and reliability of κ-FLC for the diagnosis of multiple sclerosis [7]. Currently, multiple sclerosis is identified via intrathecal oligoclonal bands (OCBs) and a positive Reiber scheme. However, several groups have reported positive κ-FLCs in MS patients using a quantitative immunoassay to examine the amount of κ-FLCs within the patient’s CSF [8,9,10].

At the basic level, the diagnostic immunoassay is ideally as sensitive as a radioimmunoassay. The immunoassay detects κ-FLCs with an antiserum specific to the free kappa chains [5]. This strategy for diagnosing multiple sclerosis has a high probability of differentiating patients with and without multiple sclerosis [11,12,13,14,15,16,17,18,19,20,21,22,23,24,25]. In this paper, a systematic literature review was performed, and essential insights into the up-to-date, state-of-the-art usage of free light chains in diagnosing multiple sclerosis are provided. Studies have advocated free light chains, especially kappa free light chains, as a practically simpler and less costly quantitative option than oligoclonal bands [26,27,28,29,30,31,32,33,34,35,36,37,38,39,40].

FLC detection is performed by an automated nephelometric test (e.g., Binding Site (Birmingham, UK), or Siemens Healthineers (Erlangen, Germany)) and/or an ELISA (e.g., Sebia(Lisses, France)). These assays are fast (<2 h) and inexpensive. In contrast, OCBs are determined by electrophoresis and subsequent immunofixation and/or immunoblotting and silver staining. The main difference is that OCBs require substantial hands-on time of a trained laboratory technician and specific laboratory equipment. Furthermore, correct assessment of the gels and/or immunoblots requires considerable experience. In comparison, the FLC assay results in a numeric value, which can be reported to the physician.

### 1.1. Background Overview of FLC Usage in MS Diagnostics

The possible assays for detecting κ-FLCs and λ-FLCs have been a primary focus in investigating the use of free light chains in CSF for examining multiple sclerosis [41,42,43,44]. Notably, the free light chain assay was initially developed by the Binding Site for multiple myeloma diagnostics. Later, the free light chain assay was utilized with CSF for the diagnosis of multiple sclerosis [41,45,46,47,48,49,50,51,52,53,54,55]. The initial group of individuals to use free light chains in MS diagnostics included Fischer, Arneth, Koehler, and Lackner (2004) [41]. The findings of their subsequent report provided substantial and preliminary insights into the possibility, reliability, and efficiency of using κ-FLCs in the diagnosis of multiple sclerosis. The vast majority of recent empirical surveys, including [51,52,53,54,55,56,57,58,59,60,61,62,63,64,65,66,67,68,69,70,71,72,73,74,75,76,77,78,79,80,81,82,83,84,85,86,87,88,89,90,91,92,93,94,95,96,97,98,99,100,101,102,103,104,105,106,107,108,109,110], have widely incorporated the essence of the abovementioned report regarding the positive implications of using free light chains, especially κ-FLCs, for MS diagnosis as well as for CIS. Subsequently, Arneth and Birklein [111] became the second group of researchers to demonstrate the use of free light chains in MS diagnostics, as evident from their 2009 study. It was only a decade later that researchers began to gain a renewed interest in the use of the free light chain assay in MS diagnosis, following a publication by Kaplan et al. (2010) [110].

### 1.2. Aim

The aim of the present study was to summarize the existing studies on the use of FLCs for MS diagnostics, which is still considered controversial in the professional world and in the literature. The present manuscript compares pro studies and con studies and aims to provide more clarity in regard to the use of FLCs in MS diagnostics.

## 2. Materials and Methods

### 2.1. Data Sources

This study entails a systematic review of prior research reports and articles to derive satisfactory conclusions. The manuscript’s structure was based on the Preferred Reporting Items for Systematic reviews and Meta-Analyses (PRISMA) model. The researcher used the QUADAS-2 tool to evaluate the risk of bias for the available references. All publications were subjected to a reference standard specific to the measure against which the free light chains were being compared. The PRISMA flow diagram is shown in Figure 1.

Figure 1 shows the PRISMA flow diagram for this review. The inclusion criteria were as follows: all studies that investigated κ-FLCs and/or κ-FLCs in CSF and inflammatory diseases and/or multiple sclerosis and/or CIS and/or conversion from CIS to MS.

A preliminary literature search was conducted to identify an additional topic of concern for the study. During the study process, a comprehensive search was performed through reputable electronic databases to identify and obtain the necessary peer-reviewed articles on randomized controlled clinical trials that best illustrate and investigate the defined hypothesis of the study. The legitimate electronic databases used in the search process included Embase, Google Scholar, Web of Science, and PubMed/MEDLINE. The literature search was limited to publications within the last 20 years, from 2002 to 2022. There was no limitation concerning the geographical boundaries of the studies/sources or authors of interest.

### 2.2. Search Strategy

The specific terms used to search the internet were multiple sclerosis, cerebrospinal fluid, lambda free light chains, kappa free light chains, and free light chains. Similarly, medical subjects or MeSH terms, such as biomarkers, immune assays, kappa and lambda isoforms, and immunoglobulin, were used to facilitate the literature search in the MEDLINE and Embase databases. The above search terms were based on the current study objectives and aims. The research team selected only articles involving clinical trials and retrospective or prospective comparative and systematic reviews focusing on the use of free light chains to diagnose multiple sclerosis and/or CIS.

### 2.3. Data Collection and Analysis Process

The findings assessed in the current meta-analysis and systematic review were the use of free light chains in diagnosing multiple sclerosis, inflammatory CNS disorders, demyelination, and/or CIS. During the literature search process, the research team reviewed the bibliography sources of each of the obtained studies to identify other relevant research reports. Identical conference and publication abstracts without full information were excluded, and the remaining articles were vetted by abstract and title before the corresponding full-text assessments. The full text of each report was autonomously analyzed by the research team members who proposed the current study. This procedure was critical in verifying the eligibility of each article for inclusion.

The inclusion criteria were as follows: studies that investigated κ-FLCs and/or λ-FLCs in CSF and inflammatory diseases and/or multiple sclerosis and/or CIS and/or conversion from CIS to MS.

## 3. Results

Approximately 2204 articles were initially obtained during the preliminary search across all four electronic databases, specifically Embase, MEDLINE, Google Scholar, and Web of Science. However, after performing the initial review of the articles using their abstracts, only 900 articles were deemed relevant to the current research hypothesis statement. Next, 670 articles that were duplicates and/or triplicates were removed, resulting in 230 nonduplicate publications. Out of these, 116 were retrieved after conducting the inclusion and exclusion criteria review. The included studies comprised 35 randomized clinical trials, 21 prospective cohort studies, 19 retrospective studies, 22 controlled clinical trials, and 13 systematic review studies.

### 3.1. Free Light Chains and Immunological Abnormalities

Of the included reviewed literature sources, 14 articles focused on examining the efficiency of using free light chains to assess immunological abnormalities. Specifically, [1,2,3,4,5,6,7,8,9,10] focused on analyzing autoinflammatory and autoimmune diseases, while [11,12,13,14] explored the importance of free light chains in diagnosing inflammatory CNS disorders and immunological deficiency syndromes.

### 3.2. Free Light Chains and Multiple Sclerosis

Of the peer-reviewed sources retrieved for the current systematic review, 116 described the study of free light chains and multiple sclerosis. The most important articles were [15,16,17,18,19,20,21,22,23,24,25,26,27,28,29,41,66,111].

### 3.3. Free Light Chains and Demyelinating Diseases

At least two reports analyzed in this case study were based on assessment of the sensitivity of free light chains in distinguishing demyelinating diseases. Essentially, the studies [10,36] offered a substantial overview regarding the application of κ-FLCs in diagnosing or detecting demyelinating diseases.

### 3.4. The Efficiency of Lambda Free Light Chains in the Diagnosis of Diseases/Multiple Sclerosis

While κ-FLCs in CSF are considered an effective alternative diagnostic approach for multiple sclerosis, λ-FLCs in CSF have received only limited attention from the scientific research community. As evident from the current systematic literature search presented in Table 1 of the appendix, few studies have focused on directly investigating the positive implications of using λ-FLCs in multiple sclerosis diagnosis in different contexts. Notably, most of the studies related to the efficiency of λ-FLCs have been generally based within the wider framework of diagnostics using more than one type of free light chain.

Of the included articles, 8 sources (Kaplan et al. [2], Lock et al. [3], Bhole et al. [5], Muchtar et al. [6], Gottenberg et al. [9]., Gurtner et al. [32], Jiang et al. [34], and Draborg et al. [37]) reported that there are relatively high concentrations of both lambda and kappa isoforms in the serum of patients with autoimmune diseases.

In particular, Kaplan et al. [2] noted that λ-FLC isoforms primarily manifest in dimeric and polymeric forms, which are usually modified under immunological conditions.

Studies by Senel et al. [12], Makshakov et al. [13], Basile et al. [14], Hampson et al. [26], and Napodano et al. [30] established that cerebrospinal fluid-based free light chains are significant disease biomarkers in individuals diagnosed with inflammatory CNS diseases such as multiple sclerosis and CIS. For instance, the experiential investigation conducted by Napodano et al. [30] indicated that lambda (λ) free light chains are low-weight proteins secreted in overabundance during the synthesis of immunoglobulins and discharged into CSF and/or the circulation depending on the localization of the inflammation. In this way, the presence of FLCs in CSF is clearly connected with plasma cell action. 

Additionally, two studies by Hoedemakers et al. [7] and Campbell et al. [8] reported that there are comparable clinical differences in specificity and sensitivity between the monoclonal lambda FLC assays and the polyclonal antibody-based lambda FLC assays used for monoclonal plasma proliferative disorder diagnosis (multiple myeloma diagnostics).

The results concerning λ-FLCs are currently more controversial than those for κ-FLCs. Several studies report a higher number of patients with positive λ-FLCs in CSF than those with κ-FLCs [66,111]. This phenomenon can be explained by the fact that λ-FLCs tend to dimerize. Subsequently, dimers will not be able to cross the CSF barrier. This effect would make λ-FLCs extremely sensitive markers of intrathecal inflammation. However, there are also a few studies that were not able to detect any λ-FLCs in most of their patients [17]. These reports can probably be explained by the fact that lambda polymers can be pulled out of the sample through high centrifugation. Therefore, the preanalytical treatment of the samples plays a definitive and important role in the value of λ-FLCs in these studies and can lead to preanalytical bias.

### 3.5. The Efficiency of Kappa Free Light Chains in Diagnosing Multiple Sclerosis

A significant number of studies have endeavored to examine the efficiency of κ-FLC measurement in the diagnosis of multiple sclerosis. Comparatively, the empirical survey outcomes were reported by Ferraro et al. [17] and Bosello et al. [18]. Basile [19], Altinier et al. [23], Zeman et al. [24], and Zeman et al. [25] demonstrated that CSF kappa free light chains are more profound intrathecal immunoglobulin markers than oligoclonal bands (OCBs). The findings were consistent with the results of studies by Nazarov et al. [15], Nazarov et al. [16], Rathbone et al. [20], and Bernardi et al. [36], which also supported the positive implications of kappa free light chains in the early diagnosis of multiple sclerosis.

As evident from the information in Table 2 and Table 3 in the appendix, it is apparent that the studies focused on different aspects of κ-FLC diagnostics. For instance, studies by Rosenstein et al. [40], Fischer et al. [41], Leurs et al. [44], Villar et al. [67], Hassan-Smith et al. [33], Süße et al. [70], Vasilj et al. [73], Voortman et al. [78], Presslauer et al. [79], Senel et al. [80], Presslauer et al. [82], Huss et al. [85], Rinker et al. [92], Nakano et al. [96], Ramsden [101], Messaoudani et al. [107] and Martins et al. [112] found that there was a higher concentration of κ-FLCs in patients with clinically validated multiple sclerosis. Overall, eight studies (Mead et al. [49], Han et al. [52], Nazarov et al. [59], Kaplan et al. [60], Vecchio et al. [62], Annunziata et al. [94], Saadeh et al. [98], Arneth et al. [111]) reported that free light chains could be effectively utilized in the diagnosis of multiple sclerosis.

Similarly, four studies provided a description of the criteria used in measuring the diagnostic outcomes with κ-FLCs [50,51,52,53,54,55,56,57,58,59,60,61,62]. Several studies reported that Reiber’s diagram can be used for accurate evaluation of κ-FLCs and the associated accuracy of MS diagnosis [66,111]. All the authors provided proof that measurement of κ-FLC concentration is less expensive and technically demanding than OCB-based protocols. Accordingly, these researchers demonstrated that immunoglobin synthesis, as a biomarker of MS, can be established with simple CSF/serum κ-FLC quotients and the given albumin CSF/serum concentration quotient (quotient diagram) [28,111]. These measurements also constitute the κ-FLC index and contribute to a possible novel diagnostic pipeline for MS. These studies identified gaps in the empirically defined κ-FLC index thresholds that vary considerably across different clinical trials [39,40,41,42,43,44].

For instance, Schwenkenbecher et al. noted that the currently applicable κ-FLC index thresholds are 3.6, 5.9, 6.07, and 12 [28]. To overcome this variability of the κ-FLC indices, Arneth and Reiber developed a more accurate approach that would increase the sensitivity of the κ-FLC measurement in the intrathecal synthesis of κ-FLC values [29,111]. Reiber first did not believe in the usefulness of quotient diagrams for κ-FLC assessment in CSF. Arneth had to explain the usefulness to him, which resulted in a series of controversial letters published in Acta Neurologica Scandinavica [66].

Today, Reiber’s hyperbolic function for the assessment of κ-FLCs in CSF has gained support from several researchers who tested the analogy in separate multicenter studies with human patients [29,111].

In general, few examinations have placed a particular focus on the prognostic value of either κ-FLCs or lambda free light chains (λ-FLCs) in MS [72,73,74,75,76]. Indeed, even the few studies that researched that relationship recorded enormous and diverse complexities. One reason for these difficulties experienced by past specialists may have resulted from assessment of free kappa light chain and/or free lambda light chain levels in the CSF alone without considering the blood-CSF barrier [72,73,74,75,76] and without using quotients. Different methods of computing the proportion of FLCs in the CSF to that in the serum prevailed; however, they did not connect their discoveries with the blending of the different MS disease courses, consequently examining each piece of information in an isolated factual manner [75,76,77,78,79]. Additionally, there were irregularities in studies on the impact of κ-FLC results on MS diagnosis in situations where clinical factors such as comorbidities, socioeconomics, and MRI factors are known to the specialist, e.g., when frozen CSF samples were used in pure laboratory studies [79].

Regardless, a couple of studies successfully established the foundation for critiquing the difference in effectiveness between FLC- and OCB-based approaches to MS diagnosis. Drawing from previous works on the topic [80,81,82,83,84,85], the authors noted that κ-FLCs and OCBs offer prognostic value in predicting clinical attacks of MS among patients, but the use of κ-FLCs offers even more clinical advantages. Berek et al. noted that the κ-FLC index is measured by means of nephelometry, which is labor-saving, reliable, easier and less costly than the conventional isoelectric assessment of OCBs [82]. The study further supported the use of the κ-FLC index because of the numerical quantifiable metric values included in its diagnostic criteria as opposed to the dichotomous “optical” values provided by the OCB tests [83]. OCB tests return only negative or positive values that are perceived by visual inspection. This assertion was supported by findings from a number of previous studies that focused on the sensitivity of free light chain tests [71,80,82,111].

Studies by Villar et al. [67], Hassan-Smith et al. [69], Presslauer et al. [82], and Süße et al. [70] included relatively small patient populations with MS in comparison to other literature sources. In contrast, studies by Christiansen et al. [50], Duranti et al. [46], and Ferraro et al. [61] included fairly large patient populations from different disease categories in their respective studies, thus enabling a possible comparison of the efficiency of κ-FLCs and λ-FLCs in MS diagnosis. The publications by Vecchio et al. [62], Villar et al. [67], Arneth et al. [66], Rathbone et al. [59], and Crespi et al. [58] demonstrated significant results with a *p* value of (*p* < 0.001). The studies included a large number of patients, which was crucial for deriving valid conclusions regarding the implications of κ-FLCs in MS diagnosis.

## 4. Discussion

There are positive implications for the use of free light chains in making a valid and reliable diagnosis of multiple sclerosis in the near future. The current systematic review included 116 peer-reviewed literature sources derived from reputable electronic databases. The included articles comprised 35 randomized clinical trials, 21 prospective cohort studies, 19 retrospective studies, 22 controlled clinical trials, and 13 systematic reviews. The vast majority of the empirical findings within the past decade concerning the study topic have revealed the potential of κ-FLC measurement in CSF for diagnosing multiple sclerosis and/or other inflammatory CNS diseases, as well as demyelinating CNS disorders, among other parameters and diagnostic procedures using free light chains. For instance, the empirical findings by Kaplan [2], Lock [3], Muchtar et al. [6], Makshakov et al. [13], and [111] indicated that the use of κ free light chains in CSF had been established to give the physician an additional tool to detect intrathecal immunoglobulin synthesis, and its outcomes and effectiveness are practically identical to those of OCB testing.

Notably, intrathecal immunoglobulin synthesis generally occurs in multiple sclerosis as well as under contagious and immunological pathological conditions that involve a humoral immune reaction [5]. Similarly, other experiential studies involving OCB testing for associated diseases, such as cardiovascular and demyelinating diseases, have reported the same outcomes using free light chains. For instance, studies by Bhole et al. [5], Hoedemakers et al. [7], and Senel et al. [12] have demonstrated that the use of free light chains, specifically kappa free light chains, provides up to 95% accuracy in the diagnosis of obstructive pulmonary disease and demyelination.

The heterogeneity in the FLC index thresholds has been noted by many scholars as a potential loophole and methodological disadvantage of nephelometry [92]. The majority of the literature that discredits this approach bases its opposition on two grounds. First, there might be disagreement over the accurate cutoff points for the κ-FLC indices [92]. The inconsistent use of the κ-FLC index values would reflect nonlinear functions that make it virtually impossible to establish its diagnostic value as a fraction [59]. Throughout the literature, this drawback seems to be the primary obstacle that has resulted in the limited use of κ-FLCs in clinical practice. However, this problem can be solved by using quotient diagrams. Second, there is limited clinical experience with κ-FLC quotient diagrams.

Regardless, further investigations into the κ-FLC measurement criteria have seen tremendous breakthroughs. The validity of “Reiber’s analogy” (using Reibers Schema or quotient diagrams for κ-FLC assessment), as proven across several multicenter studies, makes it possible in the future for the use of κ-FLCs to be integrated into clinical practice [60,61,62,63,64]. Therefore, the present review recommends simultaneous kappa FLC measurements in CSF and serum and subsequent assessment using a quotient diagram. Other findings demonstrated that the efficacy of using free light chains in diagnosing multiple sclerosis and immunological abnormalities is limited to certain factors. For example, as evident in Arrambide et al. [42], the use of κ-FLCs and λ-FLCs is still limited to CSF diagnostics and therefore also requires a lumbar puncture. 

Regarding the differences between κ-FLCs and λ-FLCs, while κ-FLCs are almost identical to OCBs in terms of clinical information, λ-FLCs seem to be elevated in CSF much more frequently and are detectable to a much greater extent. λ-FLCs also occur in the CSF in patients with other pathologies. For example, they are often detectable in the CSF in patients after strokes. This is potentially explained by the fact that lambda chains have a tendency to dimerize and to polymerize. These lambda dimers can then no longer cross the blood-CSF barrier so easily and have a significantly increased biological half-life in the CSF. As a result, very weak inflammatory events are sufficient to increase λ-FLC levels.

Based on the insights of the current study, the use of FLCs in diagnostic measures is worthwhile and relatively inexpensive. As depicted by Kaplan et al. [2,41,111], the diagnosis of multiple sclerosis using κ-FLCs is technically less costly than typical state-of-the-art diagnostics such as OCBs. In addition, using FLCs is easier and often yields faster results than OCB testing. The costs of the automated FLC tests are approximately 50% of the costs of the OCB tests. In developing countries, FLC determination might be a good alternative as it does not require much laboratory training and equipment.

The use of FLCs for MS diagnostics remains controversial. In this review, the pro studies and the con studies were compared. In summary, it can be said that the pro studies with regard to κ-FLC diagnostics in CSF predominate in number and scope. For this reason, it is to be hoped that κ-FLC diagnostics will soon become part of the standard program for MS diagnostics and for assessing the progression of MS.

## 5. Conclusions

The primary purpose of the current systematic review study was to investigate an up-to-date, state-of-the-art method for multiple sclerosis CSF diagnosis using FLCs. For this purpose, the study focused on both κ-FLCs and λ-FLCs. A total of 116 sources were reviewed in the context of the study and were limited to articles published within the last 20 years (between 2002 and 2022). Based on the insights of the resulting literature, FLCs, especially in cerebrospinal fluid diagnostics, have increasingly gained popularity, particularly in the past half-decade.

In summary, there is substantial agreement in the scientific community that the diagnostic value of κ-FLCs in CSF is almost equal to OCBs in terms of sensitivity and specificity. With regard to λ-FLCs, the literature is much more heterogeneous. While several studies report a higher sensitivity of λ-FLCs in CSF for the detection of intrathecal inflammation, others report low λ-FLC values for most of the patients investigated. It is very likely that preanalytical handling of the samples plays a large role in λ-FLC diagnostics in CSF.

The reviewed articles reported that the analysis of FLCs potentially provides up to 95% accuracy in diagnosing multiple sclerosis and other associated disorders such as CIS and intrathecal inflammation. Furthermore, the diagnosis of multiple sclerosis using FLCs is relatively fast and inexpensive in contrast to conventional state-of-the-art diagnostics, including OCBs. However, limiting factors may hinder the efficiency of FLC diagnostics, and they should be identified in the coming years.

## Figures and Tables

**Figure 1 medicina-58-01512-f001:**
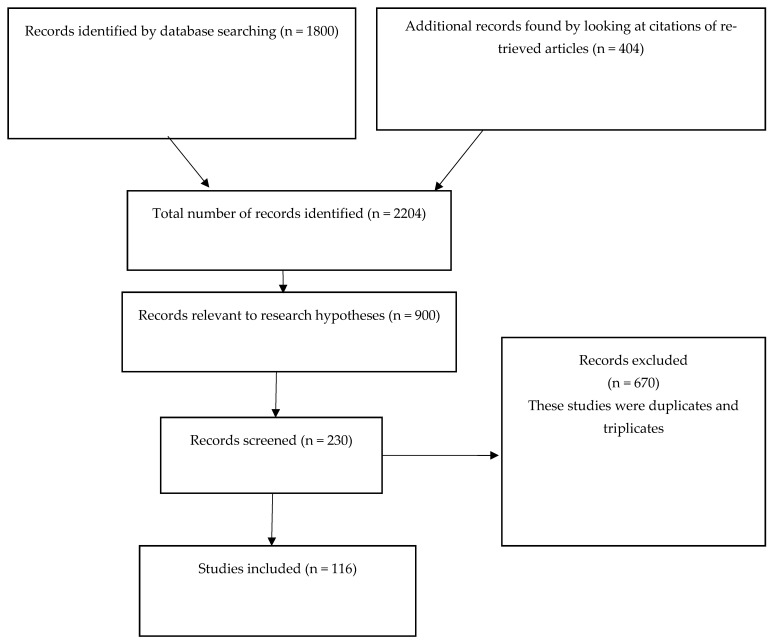
PRISMA flow diagram.

**Table 1 medicina-58-01512-t001:** Significant Findings about the Efficiency of Free Light Chain Testing in General and in Multiple Sclerosis Diagnostics.

Key Findings from the Literature	Supporting Studies	*n*	*p* Value
Free light chains are essential in altering polymorphonuclear neutrophils (PMN) functions and aiding in PMN prestimulation.	Esparvarinha et al. [1], Napodano et al. [11]	900	*p* < 0.005
High concentrations of kappa and lambda free light chains are evident in the serum of multiple myeloma patients.	Kaplan et al. [2], Lock et al. [3], Bhole et al. [5], Muchtar et al. [6], Gottenberg et al. [9], Gurtner et al. [32], Jiang et al. [34], Draborg et al. [37]	1001/2771	*p* < 0.0001
There is a comparable clinical difference in the specificity and sensitivity of diagnosing monoclonal plasma proliferative disorders between a monoclonal free light chain (FLC) assay and a polyclonal antibody-based assay.	Hoedemakers et al. [7], Campbell et al. [8]	671/890	*p* < 0.0001
Positive implications of immunoglobulin free light chains in the early diagnosis of multiple sclerosis.	4 studies, *n* = 1640 Nazarov et al. [15], Nazarov et al. [16], Rathbone et al. [20], Bernardi et al. [36]	1242/1640	*p* ≥ 0.320
Cerebrospinal fluid (CSF) kappa free light chain is a more profound and earlier intrathecal immunoglobulin marker in comparison to oligoclonal bands (OCBs).	6 studies *n* = 3054 Ferraro et al. [17], Bosello et al. [18]. Basile [19], Altinier et al. [23], Zeman et al. [24]. Zeman et al. [25]	2333/3054	*p* ≥ 5.7

PMN = polymorphonuclear neutrophils; FLC = free light chain.

**Table 2 medicina-58-01512-t002:** Key findings for Kappa Free Light Chain (κ-FLC ) Effectiveness in the Diagnosis of Multiple Sclerosis.

Key Findings from the Literature	Supporting Studies	Studies Against
κ-FLC concentrations in CSF are higher in patients with clinically validated multiple sclerosis	16 studies; *n* = 3040 (Hassan-Smith [33], Gudowska-Sawczuk et al. [27], Rosenstein et al. [40], Fischer et al. [41], Leurs et al. [47], Villar et al. [67], Süße et al. [69], Süße et al. [70], Vasilj et al. [73], Voortman et al. [78], Presslauer et al. [79], Senel et al. [80], Presslauer et al. [82], Rinker et al. [92], Nakano et al. [96], Ramsden [101], and Messaoudani et al. [107]). *n* = 2033/3040 *p* < 0.001	-
κ-FLC concentrations in CSF can be used to predict multiple sclerosis	8 studies; *n* = 1800 (Presslauer et al. [31], Mead et al. [49], Han et al. [52], Rathbone et al. [20], Kaplan et al. [60], Vecchio et al. [62], Annunziata et al. [94], Saadeh et al. [98], Bernardi et al. [36], and Abid et al. [113]). *n* = 981/1800 *p* < 0.005	-
κ-FLC index cutoff values are a novel tool in the determination of intrathecal synthesis of κ-FLCs	9 studies; *n* = 2450 (Cavalla et al. [43], Freedman et al. [55], Katzmann et al. [56], Pieri et al. [71], Arrambide et al. [81], McLean et al. [88], and Dispenzieri et al. [93]). *n* = 1721/2450 *p* < 0.005	5 studies, *n* = 1341 (Geervani et al. [74], DeCarli et al. [83], Teunissen et al. [89], Deisenhammer et al. [90,114], and Sanz Diaz et al. [91] and Magliozzi et al [115])
Reiber’s diagram provides accurate measurements of κ-FLCs and the associated accuracy of multiple sclerosis (MS ) diagnosis	5 studies, *n* =1447 (Schwenkenbecher et al. [28], Konen et al. [65], Reiber et al. [87], Arneth et al. [66,111], and Duranti et al. [46]). *n* = 911/1447 *p* < 0.013	-
κ-FLC index is a better predictor of MS than the use of CSF OCBs	11 studies, *n* = 2700 (Leurs et al. [81], Desplat-Jégo et al. [45], Duranti et al. [46], Dispenzieri et al. [93], Duell et al. [53], Bochtler et al. [91], Tintore et al. [75], Ferraro et al. [61], Gaetani et al. [100], Konen et al. [116], and Sanz et al. [91]). *n* = 2321/2700 *p* < 0.091	5 studies, *n* = 1200 (Christiansen et al. [50], Presslauer et al. [82], Crespi et al. [58], Natali et al. [84], and Joseph et al. [95])
κ-FLC concentration in CSF is the future of MS diagnosis	12 studies, *n* = 3150 (Polman et al. [39], Pröbstel et al. [68], Kyle et al. [72], Carpendale et al. [76], Abu-Izneid et al. [77], Wootla et al. [86], Smith et al. [97], Dobson et al. [102], Anagnostouli et al. [103], Valencia-Vera et al. [105], Meinl et al. [106], and Hegen et al. [114]). *n* = 2776/3150 *p* < 0.0001	-

**Table 3 medicina-58-01512-t003:** A summary of the most important studies on the role of free light chains in MS diagnosis.

Study	Study Question/Hypothesis	*n*	*p*-Value	Reported Results
Leurs et al. 2020 [44]	Can kappa free light chain (κ-FLC) and lambda free light chain (λ-FLC) indices serve as diagnostic biomarkers in multiple sclerosis?	745	*p* < 0.001	Compared with OCBs, the κ-FLC index is more sensitive but less specific for diagnosing CIS/MS.
Christiansen et al. 2018 [50]	Comparative diagnostic performance of CSF FLC with OCB and Immunoglobulin G (IgG) index.	96/230	*p* < 0.094	Using only the absolute concentration of CSF-kappa is a logistic advantage in clinical laboratories.
Crespi et al. 2019 [58]	Is the κ-FLC index a reliable marker of intrathecal synthesis and an alternative to the IgG index in multiple sclerosis diagnostic work-up?	385	*p* < 0.0001	Results confirmed the previous proposal to use the κ-FLC index as a highly sensitive and easy-to-detect first-line marker in CSF analysis for intrathecal synthesis.
Rathbone et al. 2018 [20]	Do free light chains (FLCs) as biomarkers for confirming a diagnosis of MS show greater sensitivity and specificity than OCBs?	43	*p* < 0.026	CSF immunoglobulin κ: λ ratios, determined at the time of diagnostic lumbar puncture, predict MS disease progression and may therefore be useful prognostic markers for early therapeutic stratification.
Vecchio et al. 2020 [62]	What is the role of κ-FLCs in the diagnostic work-up for MS?	406	*p* < 0.001	κ-FLCs provided high sensitivity and decent specificity for MS diagnosis.
Arneth et al. 2009 [66,111]	Immunoglobulin free light chain concentrations measured in the CSF of patients with neurological disorders.	20	*p* < 0.001	The high sensitivity of lambda light chains for the detection of intrathecal immunoglobulin synthesis may be of benefit in establishing clinical diagnoses.
Villar et al. 2012 [67]	What is the accuracy of CSF κ-FLC measurement to predict the conversion of CIS patients to MS?	133/374	*p* < 0.001	High CSF κ-FLC concentration accurately predicts the conversion of CIS patients to MS.
Süße et al. 2020 [70]	What is the application and interpretation of κ-FLC data in quotient diagrams with a hyperbolic reference range?	98/400	*p* < 0.001	The evaluation of κ-FLC with a hyperbolic reference range in quotient diagrams is superior to other analytical methods, such as the linear κ-FLC index.
Voortman et al. 2016 [78]	What is the prognostic value of κ-FLC in OCB-positive patients with clinically isolated syndrome (CIS) suggestive of MS and early MS?	48/61	*p* < 0.05	Increased intrathecal synthesis of κ-FLC in CIS/MS supports its diagnostic contribution.
Presslauer et al. 2016 [82]	What is the diagnostic accuracy of intrathecal κ-FLC synthesis?	70/438	*p* ≥ 5.9	Findings support the diagnostic value of intrathecal κ-FLC synthesis in CIS and MS patients and demonstrate a valid, easier, and rater-independent alternative to OCB detection.
Ferraro et al. 2020 [61]	What is the diagnostic accuracy of the κ-FLC index in comparison with OCB detection in predicting MS?	84/540	*p* ≥5.8	The κ-FLC index has a slightly higher sensitivity and lower specificity than CSF OCB, and both markers supply the clinician with useful, complementary information.
Saadeh et al. 2021 [98]	What are the reference values for FLC measures? What is their accuracy with regard to the diagnosis of MS?	70/1224	*p* ≥ 5.9	CSF κ-FLCs may not replace OCBs, but they may support diagnosis in MS as a quantitative parameter.
Duranti et al. 2013 [46]	Is the κ-FLC index more accurate than other parameters?	33/80	*p* < 0.001	Nephelometric assay for κ-FLCs in CSF reliably detects intrathecal immunoglobulin synthesis and discriminates multiple sclerosis patients.
Valencia-Vera et al. 2018 [105]	What is the diagnostic value of κ-FLC and its inclusion in a procedure algorithm along with OCB interpretation?	123	*p* < 0.001	κ-FLC determination is rapid and automatized, but it has no higher sensitivity or specificity than OCB in MS diagnosis.
Süße et al. 2018 [109]	Can the determination of the κ-FLC index be used to predict the presence of OCBs?	46/295	*p* < 0.86	Determination of the κ-FLC index provided a quantitative parameter that could be used as an initial diagnostic step in inflammatory central nervous system disorders before measuring OCBs.

## Data Availability

All relevant data is available in the manuscripts tables.

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
