# Peer review of "The Use of Kappa Free Light Chains to Diagnose Multiple Sclerosis"

_medicina, 2022, doi:10.3390/medicina58111512_

Round 1

Reviewer 1 Report

The authors should describe more clearly the usefulness of FLC in the diagnosis and management of multiple sclerosis.

Tables 1-3 are complicated and should be more simple and concise.

Tables show the results, and should be moved to the result section.

The results are descriptive, and the findings should be summarized.

The authors should discuss about different findings on kappa and lambda FLC, and the possible mechanisms of different regulations.

The authors should cite the references that show recently usefulness of polyclonal FLCs for the diagnosis of other diseases such as cardiovascular diseases, diabetes etc.

Author Response

Point-to-Point Answering Letter to Reviewer 1

Comments and Suggestions for Authors

The authors should describe more clearly the usefulness of FLC in the diagnosis and management of multiple sclerosis.

A: Thank you for this comment. The following sentences have been added to the introduction to address this point:

Intrathecal kappa‑FLC (k-FLC) synthesis has similar diagnostic accuracy to the well-established method of CSF-restricted oligoclonal bands (OCB) to identify patients with MS, and recent studies even report its value for the prediction of early MS disease activity. Furthermore, detection of k-FLC has significant methodological advantages in comparison to OCB detection.

Tables 1-3 are complicated and should be more simple and concise.

A: Thank you for this comment. In the revised version, Table 1 has been simplified by removing the last column. Table 2 has also been simplified.

Tables show the results, and should be moved to the result section.

A: Thank you for this comment. In the revised version, the tables have been moved to the results section.

The results are descriptive, and the findings should be summarized.

A: Thank you very much for this comment. The following summary was added at the end of the discussion section:

The use of FLCs for MS diagnostics remains controversial. In this review, the pro studies and the con studies were compared. In summary, it can be said that the pro studies with regard to k-FLC diagnostics in CSF predominate in number and scope. For this reason, it is to be hoped that k-FLC diagnostics will soon become part of the standard program for MS diagnostics and for assessing the progression of MS.

The following information was added to the conclusion section:

In summary, there is substantial agreement in the scientific community that the diagnostic value of k-FLCs in CSF is almost equal to OCBs in terms of sensitivity and specificity. With regard to l-FLCs, the literature is much more heterogeneous. While several studies report a higher sensitivity of l-FLCs in CSF for the detection of intrathecal inflammation, others report low l-FLC values for most of the patients investigated. It is very likely that preanalytical handling of the samples plays a large role in l-FLC diagnostics in CSF.

The authors should discuss about different findings on kappa and lambda FLC, and the possible mechanisms of different regulations.

A: Thank you very much for this comment and for this important point. The following sentences were added to the discussion section:  

Regarding the differences between k-FLCs and l-FLCs, while k-FLCs are almost identical to OCBs in terms of clinical information, l-FLCs seem to be elevated in CSF much more frequently and are detectable to a much greater extent. l-FLCs also occur in the CSF in patients with other pathologies. For example, they are often detectable in the CSF in patients after strokes. This is potentially explained by the fact that lambda chains have a tendency to dimerize and to polymerize. These lambda dimers can then no longer cross the blood-CSF barrier so easily and have a significantly increased biological half-life in the CSF. As a result, very weak inflammatory events are sufficient to increase l-FLC levels.

The authors should cite the references that show recently usefulness of polyclonal FLCs for the diagnosis of other diseases such as cardiovascular diseases, diabetes etc.

A: Thank you for this comment. Since the review is about FLCs in CFS in MS patients, we did not want to spend too much time discussing other fields. However, cardiovascular has been added to the following sentence in the discussion section:

The vast majority of the empirical findings within the past decade concerning the study topic have revealed the potential of k-FLC measurement in CSF for diagnosing multiple sclerosis and/or other inflammatory CNS diseases, as well as demyelinating CNS disorders, among other parameters and diagnostic procedures using free light chains.

Reviewer 2 Report

Your paper represents a comprehensive review of publications supporting the use of Kappa Free Light Chains to Diagnose Multiple Sclerosis. Your review and effort should be commended. This reviewer is left however with some doubts that I believe should be clarified. 

1. In line 25 (Abstract) the statement "...diagnosis of various scleroses..." Please clarify this comment. 

2. One of the main conclusions your review draws is that determination in CSF of free light chains is faster and less expensive in comparison with the current methods and costs involved in the identification of Oligoclonal Bands (OGBs). This beneficial difference is not clear to the reader.. How faster (time wise?) and how less expensive percentage wise? or can you provide an estimation in international currencies? i.e. USD or Euros.  In many third world countries OGBs are not performed precisely due to either laboratory cost or lack of methodology.

3. Also, it is not clear in the flow of the text, if the review addresses the determination of Kappa Light Chains in serum simultaneously with CSF analysis, and how absence of unique KLC in CSF (in other words also present in serum) should be interpreted. 

Author Response

Point-to-Point Answering Letter to Reviewer 2

Comments and Suggestions for Authors

Your paper represents a comprehensive review of publications supporting the use of Kappa Free Light Chains to Diagnose Multiple Sclerosis. Your review and effort should be commended. This reviewer is left however with some doubts that I believe should be clarified.

A: Thank you very much for the recommendation. Yes, I would like to clarify your points.

  1. In line 25 (Abstract) the statement "...diagnosis of various scleroses..." Please clarify this comment.

A: Thank you for this comment. The sentence in question has been rephrased, and “various sclerosis” has been replaced by “CIS and multiple sclerosis”.

  1. One of the main conclusions your review draws is that determination in CSF of free light chains is faster and less expensive in comparison with the current methods and costs involved in the identification of Oligoclonal Bands (OGBs). This beneficial difference is not clear to the reader. How faster (time wise?) and how less expensive percentage wise? or can you provide an estimation in international currencies? i.e. USD or Euros.  In many third world countries OGBs are not performed precisely due to either laboratory cost or lack of methodology.

A: We thank the reviewer for this question. The following information has been added to the introduction:

FLC detection is performed by an automated nephelometric test (e.g. Binding Site,or Siemens) and/or an ELISA (Sebia). These assays are fast (<2 hours) and inexpensive. In contrast, OCBs are determined by electrophoresis and subsequent immunofixation and/or immunoblotting and silver staining. The main difference is that OCBs require substantial hands-on time of a trained laboratory technician and specific laboratory equipment. Furthermore, correct assessment of the gels and/or immunoblots requires considerable experience. In comparison, the FLC assay results in a numeric value, which can be reported to the physician.

The following information has been added to the discussion section:

The costs of the automated FLC tests are approximately 50% of the costs of the OCB tests. In developing countries, FLC determination might be a good alternative, as it does not require much laboratory training and equipment.

  1. Also, it is not clear in the flow of the text, if the review addresses the determination of Kappa Light Chains in serum simultaneously with CSF analysis, and how absence of unique KLC in CSF (in other words also present in serum) should be interpreted.

A: We thank the reviewer for this point. If kappa FLCs are present only in serum and not in CSF, then no intrathecal inflammation (and most likely no MS) is present. This constellation suggests the presence of another source of FLCs, e.g., a multiple myeloma (monoclonal gammopathy). The following sentence has been added to the discussion section:

Therefore, the present review recommends simultaneous kappa FLC measurements in CSF and serum and subsequent assessment using a quotient diagram

Reviewer 3 Report

This is a review article about the use of light chains in the diagnosis of MS. The methodology is clearly described and the authors have included recent studies.. Therefore , although this topic has been addressed many times, this review may serve as an updated paper about their use in MS. In the entire manuscript, there are several grammar and syntax errors , that render this mansucript difficult to read. Besides, full revision of the text, there as some specific comments that may help imporve the manuscript: 

Abstract : 

1. the phrase "t is often presumed that free light chains, particularly free kappa and lambda light chains, are practical  and associated with a higher percentage of obtaining positive results "   . Practical to what ? positive results for what? This sentence needs rephrasing.

2. "various scleroses" , to what scleroses do the authors refer to ? Please rephrase accordingly.

Introduction :

1. " infection-altering treatments" , do the authors mean disease modifying treatments ?  In addition the whole paragraph should be revised, since due to grammar and syntax errors is incomprehensible.

2. "Based on the pre-existing reports, multiple sclerosis is identified via
a quantitative immunoassay performed to examine the amount of free kappa light chains" This is rather a proposal. Diagnostic criteria for MS do not include light chains up to now.

Methods : a flow chart with the inclusion of the studies is needed.
within the patient’s CSF

Discussion : Since numerous metanalyses and reviews have investigated the role of the light chains in MS, and have been taken into account in this paper, the authors are suggested to include a statement of the importance of this review. What new has this review to offer in the litterature? Perhaps an updated review with the inclusion of more recent studies?

Author Response

Point-to-Point Answering Letter to Reviewer 3

Comments and Suggestions for Authors

This is a review article about the use of light chains in the diagnosis of MS. The methodology is clearly described and the authors have included recent studies. Therefore, although this topic has been addressed many times, this review may serve as an updated paper about their use in MS. In the entire manuscript, there are several grammar and syntax errors, that render this manuscript difficult to read. Besides, full revision of the text, there as some specific comments that may help improve the manuscript:

A: Thank you for this comment and for your help in improving this manuscript. The use of kappa FLCs (and even more so for lambda FLCs) in MS diagnosis is still very controversial, as discussed in the literature. There are many pro papers but also several con papers. The present manuscript tries to give a picture of the current situation.

Abstract:

  1. the phrase "t is often presumed that free light chains, particularly free kappa and lambda light chains, are practical and associated with a higher percentage of obtaining positive results "   . Practical to what? positive results for what? This sentence needs rephrasing.

A: Thank you for this comment. The sentence in question has been rephrased:

It is often presumed that free light chains, particularly kappa and lambda free light chains, are of practical use and are associated with a higher probability of obtaining positive results compared to oligoclonal bands.

  1. "various scleroses“, to what scleroses do the authors refer to ? Please rephrase accordingly.

A: Thank you for this comment. The sentence in question has been rephrased: “various sclerosis” has been replaced by “CIS and multiple sclerosis”.

Introduction:

  1. " infection-altering treatments“, do the authors mean disease modifying treatments ?  In addition, the whole paragraph should be revised, since due to grammar and syntax errors is incomprehensible.

A: Thank you for this comment. The sentence and paragraph in question have been rephrased, and “infection-altering treatments” has been replaced by “disease-modifying treatments”.

  1. "Based on the pre-existing reports, multiple sclerosis is identified via
    a quantitative immunoassay performed to examine the amount of free kappa light chains" This is rather a proposal. Diagnostic criteria for MS do not include light chains up to now.

A: Thank you for this comment. The sentence and paragraph in question have been rephrased:

Until now, the diagnostic criteria for MS have not included FLCs. However, in the future, k-FLC testing could possibly support or even replace OCB testing in CSF.

Methods: a flow chart with the inclusion of the studies is needed.
within the patient’s CSF

A: Thank you for this comment. A short flow-chart has been added.

Discussion: Since numerous meta-analyses and reviews have investigated the role of the light chains in MS, and have been taken into account in this paper, the authors are suggested to include a statement of the importance of this review. What new has this review to offer in the literature? Perhaps an updated review with the inclusion of more recent studies?

A: Thank you for this point. As described above, the purpose of this manuscript is to summarize the studies already available. The use of FLCs for MS diagnostics is still very controversial in the professional world and in the literature. The present manuscript wants to compare pro studies and con studies and thus provide more clarity in regard to the use of FLCs in MS diagnosis. It should help both insiders and beginners. It can be a starting point that does not simply repeat the existing literature but rather presents a comprehensive picture of the current situation. In summary, the pro studies in favor of using kappa FLCs clearly predominate. This summary should also be conveyed to the reader. Therefore, the following chapter was added at the beginning:

1.2 Aim

The aim of the present study was to summarize the existing studies on the use of FLCs for MS diagnostics, which is still considered controversial in the professional world and in the literature. The present manuscript compares pro studies and con studies and aims to provide more clarity in regard to the use of FLCs in MS diagnostics.

The following summary was added to the conclusion section: In summary, there is substantial agreement in the scientific community that the diagnostic value of k-FLCs in CSF is almost equal to OCBs in terms of sensitivity and specificity. With regard to l-FLCs, the literature is much more heterogeneous. While several studies report a higher sensitivity of l-FLCs in CSF for the detection of intrathecal inflammation, others report low l-FLC values for most of the patients investigated. It is very likely that preanalytical handling of the samples plays a large role in l-FLC diagnostics in CSF.

Round 2

Reviewer 1 Report

There are many duplicate description of the authors and numbers of the references in the tables and text. The authors' names which are described in the tables should be deleted in the text and cited only reference numbers.

Author Response

There are many duplicate description of the authors and numbers of the references in the tables and text. The authors' names which are described in the tables should be deleted in the text and cited only reference numbers.

Dear Reviewer,

thanks for this comment. The authors names have been omitted from the manuscript text when numbers are presented where possible, in the revised version of the manuscript.

Thanks.

Reviewer 3 Report

tha authors adequately addressed all my comments

Author Response

tha authors adequately addressed all my comments

A: Dear reviewer, thanks a lot for this positive comment.